# INITIALIZED EQUILIBRIUM PROPAGATION FOR BACKPROP-FREE TRAINING

**Peter O'Connor, Efstratios Gavves, Max Welling**
QUVA Lab
University of Amsterdam
Amsterdam, Netherlands
`peter.ed.oconnor@gmail.com,egavves@uva.nl,m.welling@uva.nl`

## ABSTRACT

Deep neural networks are almost universally trained with reverse-mode automatic differentiation (a.k.a. backpropagation). Biological networks, on the other hand, appear to lack any mechanism for sending gradients back to their input neurons, and thus cannot be learning in this way. In response to this, Scellier & Bengio (2017) proposed *Equilibrium Propagation* - a method for gradient-based training of neural networks which uses only local learning rules and, crucially, does not rely on neurons having a mechanism for back-propagating an error gradient. Equilibrium propagation, however, has a major practical limitation: inference involves doing an iterative optimization of neural activations to find a fixed-point, and the number of steps required to closely approximate this fixed point scales poorly with the depth of the network. In response to this problem, we propose *Initialized Equilibrium Propagation*, which trains a feedforward network to initialize the iterative inference procedure for Equilibrium propagation. This feed-forward network learns to approximate the state of the fixed-point using a local learning rule. After training, we can simply use this initializing network for inference, resulting in a learned feedforward network. Our experiments show that this network appears to work as well or better than the original version of Equilibrium propagation while requiring fewer steps to converge. This shows how we might go about training deep networks without using backpropagation.

## 1 INTRODUCTION

Deep neural networks are almost always trained with gradient descent, and gradients are almost always computed with backpropagation. For those interested in understanding the working of the brain in the context of machine learning, it is therefore distressing that biological neurons appear not to send signals backwards.

Biological neurons communicate by sending a sequence of pulses to downstream neurons along a one-way signaling pathway called an "axon". If neurons were doing backpropagation, one would expect a secondary signalling pathway wherein gradient signals travel backwards along axons. This appears not to exist, so it seems that biological neurons cannot be doing backpropagation.

Moreover, backpropagation may not be the ideal learning algorithm for efficient implementation in hardware, because it involves buffering activations for each layer until an error gradient returns. This requirement becomes especially onerous when we wish to backpropagate through many steps of time, or through many layers of depth. For these reasons, researchers are looking into other means of neural *credit assignment* - mechanisms for generating useful learning signals without doing backpropagation.

Recently, Scellier & Bengio (2017) proposed a novel algorithm called *Equilibrium Propagation*, which enables the computation of parameter gradients in a deep neural network without backpropagation. Equilibrium Propagation defines a neural network as a dynamical system, whose dynamics

---

[0] Code available at `https://github.com/QUVA-Lab/init-eqprop`

follow the negative-gradient of an energy function. The "prediction" of this network is the fixed-point of the dynamics - the point at which the system settles to a local minimum energy given the input, and ceases to change. Because of this inference scheme, Equilibrium Propagation is impractically slow for large networks - the network has to iteratively converge to a fixed point at every training iteration.

In this work, we take inspiration from Hinton et al. (2015) and distill knowledge from a slow, energy based *equilibrating network* into a fast *feedforward network* by training the feedforward network to predict the fixed-points of the equilibrating network with a local loss. At the end of training, we can then discard the equilibrating network and simply use our feedforward network for test-time inference. We thus have a way to train a feedforward network without backpropagation. The resulting architecture loosely resembles a Conditional Generative Adversarial Network (Mirza & Osindero, 2014), where the feedforward network produces a network state which is evaluated by the energy-based equilibrating network.

To aid the reader, this paper contains a glossary of symbols in Appendix A.

## 2 METHODS

### 2.1 BACKGROUND: EQUILIBRIUM PROPAGATION

Equilibrium Propagation (Scellier & Bengio, 2017) is a method for training an Energy-Based Model Hopfield (1984) for classification. The network performs inference by iteratively converging to a fixed-point, conditioned on the input data, and taking the state of the output neurons at the fixed point to be the output of the network. The network's dynamics are defined by an energy function over neuron states $s$ and parameters $\theta = (w, b)$:

$$E_\theta(s, x) = \frac{1}{2} \sum_{i \in \mathcal{S}} s_i^2 - \sum_{i \in S} b_i \rho(s_i) - \sum_{j \in \mathcal{S}, i \in \alpha_j \cap \mathcal{S}} w_{ij} \rho(s_i) \rho(s_j) - \sum_{j \in \mathcal{S}, i \in \alpha_j \cap \mathcal{I}} x_i w_{ij} \rho(s_j) \quad (1)$$

Where $\mathcal{I}$ is the set of input neuron indices, $\mathcal{S}$ is the set of non-input neuron indices; $s \in \mathbb{R}^{|\mathcal{S}|}$ is the vector of neuron states; where $\alpha_j \subset \{\mathcal{I} \cup \mathcal{S}\}$ is the set of neurons connected to neuron $j$; $x$ denotes the input vector; and $\rho$ is some nonlinearity; $w$ is a weight matrix with a symmetric constraint: $w_{ij} = w_{ji}$[1], and entries only defined for $\{w_{ij} : i \in \alpha_j\}$ The state-dynamics for non-input neurons, derived from Equation 1, are:

$$\frac{\partial s_j}{\partial t} = -\frac{\partial E_\theta(s, x)}{\partial s_j} = -s_j + \rho'(s_j) \left( b_j + \sum_{j \in \mathcal{S}, i \in \alpha_j \cap \mathcal{S}} w_{ij} \rho(s_i) + \sum_{j \in \mathcal{S}, i \in \alpha_j \cap \mathcal{I}} w_{ij} x_i \right) \forall j \in \mathcal{S} \tag{2}$$

The network is trained using a two-phase procedure, with a negative and then a positive phase. In the negative phase, the network is allowed to settle to an energy minimum $s^- := \arg\min_s E_\theta(s, x)$ conditioned on a minibatch of input data $x$. In the positive phase, a target $y$ is introduced, and the energy function is augmented to "perturb" the fixed-point of the state towards the target with a "clamping factor" $\beta$: $E_\theta^\beta(s, x, y) = E_\theta(s, x) + \beta C(s_\mathcal{O}, y)$, where $\beta$ is a small scalar and $C(s_\mathcal{O}, y)$ is a loss defined between the output neurons in the network and the target $y$ (we use $C(s_\mathcal{O}, y) = \|s_\mathcal{O} - y\|_2^2$). The network is allowed to settle to the perturbed state $s^+ := \arg\min_s (E^\beta(s, x, y))$.

Finally, the parameters of the network are learned based on a contrastive loss between the negative-phase and positive-phase energy, which can be shown to be proportional to the gradient of the output loss in the limit of $\beta \to 0$:

---

[1]However as described in Scellier et al. (2018), the symmetry requirement can be relaxed without significantly impacting performance. In this case there is no energy function, but one simply defines the network in terms of the state dynamics. The network nevertheless seems to learn to settle to fixed-points rather than falling into limit cycles or chaotic dynamics. The reason for this phenomenon is not well understood.

$$\Delta\theta = -\frac{\eta}{\beta} \left( \frac{\partial E_\theta(s^+, x)}{\partial \theta} - \frac{\partial E_\theta(s^-, x)}{\partial \theta} \right) \propto -\frac{\partial C(s_\mathcal{O}^-, y)}{\partial \theta} \tag{3}$$

Where $\eta$ is some learning rate; $\mathcal{O} \subseteq \mathcal{S}$ is the subset of output neurons. This results in a local learning rule, where parameter changes only depend on the activities of the pre- and post-synaptic neurons:

$$\Delta w_{ij} = \frac{\eta}{\beta} \left( \rho(s_i^+)\rho(s_j^+) - \rho(s_i^-)\rho(s_j^-) \right) \tag{4}$$

$$\Delta b_i = \frac{\eta}{\beta} \left( \rho(s_i^+) - \rho(s_i^-) \right) \tag{5}$$

Intuitively, the algorithm works by adjusting $\theta$ to pull $\arg\min_s E_\theta(s, x)$ closer to $\arg\min_s E_\theta^\beta(s, x, y)$ so that the network will gradually learn to naturally minimize the output loss.

Because inference involves an iterative settling process, it is an undesirably slow process in Equilibrium propagation. In their experiments, Scellier & Bengio (2017) indicate that the number of settling steps required scales super-linearly with the number of layers. This points to an obvious need for a fast inference scheme.

## 2.2   ADDING AN INITIALIZATION NETWORK

We propose training a feedforward network $f_\phi(x) \to s^f \in \mathbb{R}^{|\mathcal{S}|}$ to predict the fixed-point of the equilibrating network. This allows the feedforward network to achieve two things: First, it initializes the state of the equilibrating network, so that the settling process starts in the right regime. Second, the feedforward network can be used to perform inference at test-time, since it learns to approximate the minimal-energy state of the equilibrating network, which corresponds to the prediction. $f_\phi(x)$ is defined as follows:

$$
\begin{aligned}
f_\phi(x) &:= (s_j^f : j \in \mathcal{S}) && \in \mathbb{R}^{|\mathcal{S}|} \\
s_j^f &:= \rho\left( \left( \sum_{i \in \alpha_j^f \cap \mathcal{S}} \omega_{ij} s_i^f \right) + \left( \sum_{i \in \alpha_j^f \cap \mathcal{I}} \omega_{ij} x_i \right) + c_j \right) \in \mathbb{R}
\end{aligned}
\tag{6}
$$

Where $\alpha_j^f = (i : (i \in \alpha_j) \wedge (i < j))$ is the set of *feedforward* connections to neuron $j$ (which is a subset of $\alpha_j$ - the full set of connections to neuron $j$ from the equilibrium network from Equation 1); $\phi = (\omega, c)$ is the set of parameters of the feedforward network. This feedforward network produces the initial state of the negative phase of equilibrium propagation network, given the input data - i.e., instead of starting at a zero-state, the equilibrium-propagation network is initialized in a state $s^f := f_\phi(x)$. We train the parameters $\phi$ to approximate the minimal energy state $s^-$ of the equilibrating network [2]. In other words, we seek:

$$\phi^* := \arg\min_\phi \mathcal{L}(s^f, s^-) \tag{7}$$

$$\mathcal{L}(s^f, s^-) := \sum_{i \in \mathcal{S}} \mathcal{L}_i(s_i^f, s_i^-) := \sum_{i \in \mathcal{S}} (s_i^f - s_i^-)^2 \tag{8}$$

The derivative of the forward parameters of the $i$'th neuron, $\phi_i = (\omega_{\alpha_i, i}, c_i)$, can be expanded as:

_______________________

[2]We could also minimize the distance with $s^+$, but found experimentally that this actually works slightly worse than $s^-$. We believe that this is because equilibrium propagation depends on $s^-$ being very close to a true minimum of the energy function, and so initializing the negative phase to $s^f \approx s^-$ will lead to better gradient computations than when we initialize the negative phase to $s^f \approx x^+$

$$\frac{\partial \mathcal{L}}{\partial \phi_i} := \sum_{j \in \mathcal{S}} \frac{\partial \mathcal{L}_j(s_j^f, s_j^-)}{\partial \phi_i} = \overbrace{\frac{\partial \mathcal{L}_i}{\partial s_i^f} \frac{\partial s_i^f}{\partial \phi_i}}^{local} + \overbrace{\sum_{j>i} \frac{\partial \mathcal{L}_j}{\partial s_j^f} \frac{\partial s_j^f}{\partial s_i^f} \frac{\partial s_i^f}{\partial \phi_i}}^{distant} \tag{9}$$

The *distant* term is problematic, because computing $\frac{\partial s_j^f}{\partial s_i^f}$ would require backpropagation, and the entire purpose of this exercise is to train a neural network without backpropagation. However, we find that only optimizing the local term $\frac{\partial \mathcal{L}_i}{\partial \phi_i}$ does not noticeably harm performance. In Section 2.4 we go into more detail on why it appears to be sufficient to minimize local losses.

Over the course of training, parameters $\phi$ will learn until our feedforward network is a good predictor of the minimal-energy state of the equilibrating network. This feedforward network can then be used to do inference: we simply take the state of the output neurons to be our prediction of the target data. The full training procedure is outlined in Algorithm 1. At the end of training, inference can be done either by taking the output activations from the forward pass of the inference network $f_\phi$ (Algorithm 2), or by initializing with a forward pass and then iteratively minimizing the energy (Algorithm 3). Experiments in Section 3 indicate that the forward pass performs just as well as the full energy minimization.

---

**Algorithm 1** Training

1: **Input:** Dataset $(x, y)$, Step Size $\epsilon$, Learning Rate $\eta$, Network Architecture $\alpha$, Number of negative-phase steps $T^-$, Number of positive-phase steps $T^+$
2: $\phi \leftarrow$ InitializeFeedforwardParameters($\alpha$)
3: $\theta \leftarrow$ InitializeEquilibriumParameters($\alpha$)
4: **while** not converged **do**
5: $\quad x_m, y_m \rightarrow$ SampleMinibatch($x, y$)
6: $\quad s \leftarrow s^f \leftarrow f_\phi(x_m)$
7: $\quad$ **for** $t \in 1..T^-$ **do** # Neg. Phase
8: $\quad\quad s \leftarrow s - \epsilon \frac{\partial E_\theta(s, x_m)}{\partial s}$
9: $\quad s^- \leftarrow s$
10: $\quad$ **for** $t \in 1..T^+$ **do** # Pos. Phase
11: $\quad\quad s \leftarrow s - \epsilon \frac{\partial E_\theta^\beta(s, x_m, y_m)}{\partial s}$
12: $\quad s^+ \leftarrow s$
13: $\quad \theta \leftarrow \theta - \frac{\eta}{\beta} \left( \frac{\partial E_\theta(s^+, x)}{\partial \theta} - \frac{\partial E_\theta(s^-, x)}{\partial \theta} \right)$
14: $\quad \phi_i \leftarrow \phi_i - \eta \frac{\partial \mathcal{L}_i(s_i^f, s_i^-)}{\partial \phi_i} \forall i$
15: **Return:** $\phi, \theta$ # Parameters

---

**Algorithm 2** Feedforward Inference

1: **Input:** Input Data $x$, Inference Parameters $\phi$
2: $s \leftarrow f_\phi(x)$
3: **return** $(s_i : i \in \mathcal{O})$ # Output unit states

---

**Algorithm 3** Iterative Inference

1: **Input:** Input Data $x$, Initialization Parameters $\phi$, Equilibrating Parameters $\theta$, Number of Negative Steps $T^-$
2: $s \leftarrow f_\phi(x)$
3: **for** $t \in 1..T^-$ **do** # Neg. Phase
4: $\quad s \leftarrow s - \epsilon \frac{\partial E_\theta(s, x_m)}{\partial s}$
5: **return** $(s_i : i \in \mathcal{O})$ # Output unit states

---

### 2.3 INCLUDING THE FORWARD STATES IN THE ENERGY FUNCTION

The fixed point $s^-$ of the equilibrating network is a nonlinear function of $x$, whose value is computed by iterative bottom-up and top-down inference using all of the parameters $\theta$. The initial state $s^f$, by contrast, is generated in a single forward pass, meaning that the function relating $s_j^f$ to its direct inputs $s_{\alpha_j}^f \in \mathbb{R}^{|\alpha_j|}$ is constrained to the form of Equation 6. Because of this, the computation resulting in $s^-$ may be more *flexible* than that of the forward pass, so it is possible for the equilibrating network to create targets that are not *achievable* by the neurons in the feedforward network. This is similar to the notion of an "amortization gap" in variational inference, and we discuss this connection more in Section 4.2.

Neurons in the feedforward network simply learn a linear mapping from the previous layer's activations to the targets provided by the equilibrating network. In order to encourage the equilibrating

network to stay in the regime that is reachable by the forward network, we can add a loss encouraging the fixed points to stay in the regime of the forward pass.

$$E_\theta^\lambda(s, x) = E_\theta(s, x) + \lambda \sum_{j \in \mathcal{S}} (s_j^f - s_j)^2 \tag{10}$$

Where $\lambda$ is a hyperparameter which brings the fixed-points of the equilibrating network closer to the states of the forward pass, and encourages the network to optimize the energy landscape in the region reachable by the forward network. Of course this may reduce the effective capacity of the equilibrating network, but if our goal is only to train the feedforward network, this does not matter. This trick has a secondary benefit: It allows faster convergence in the negative phase by pulling the minimum of $E_\theta^\lambda(s, x)$ closer to the feedforward prediction, so we can learn with fewer convergence steps. It can however, cause instabilities when set too high. We investigate the effect of different values of $\lambda$ with experiments in Appendix D.

## 2.4 WHY THE LOCAL LOSS IS SUFFICIENT: GRADIENT ALIGNMENT

In Equation 9 we decompose the loss-gradient of parameters $\phi$ into a local and a global component. Empirically (see Figures 1, 3), we find that using the local loss and simply ignoring the global loss led to equally good convergence. To understand why this is the case, let use consider a problem where we learn the mapping from an input $x$ to a set of neuron-wise targets: $s^*$. Assume these targets are generated by some (unknown) set of ideal parameters $\phi^*$, so that $s^* = f_{\phi^*}(x)$. To illustrate, we consider a two layer network with $\phi = (w_1, w_2)$ and $\phi^* = (w_1^*, w_2^*)$:

$$
\begin{aligned}
s_1 &= \rho(xw_1) & s_1^* &= \rho(xw_1^*) & \mathcal{L}_1 &= \|s_1 - s_1^*\|_2^2 \\
s_2 &= \rho(s_1 w_2) & s_2^* &= \rho(s_1^* w_2^*) & \mathcal{L}_2 &= \|s_2 - s_2^*\|_2^2
\end{aligned} \tag{11}
$$

It may come as a surprise that when $\phi$ is in the neighbourhood of the ideal parameters $\phi^*$, the cosine similarity between the *local* and *distant* gradients: $S\left(\frac{\partial \mathcal{L}_1}{\partial w_1}, \frac{\partial \mathcal{L}_2}{\partial w_1}\right)$ is almost always positive, i.e. the local and distant gradients tend to be aligned. This is a pleasant surprise because it means the local loss will tend to guide us in the right direction. The reason becomes apparent when we define $\Delta w := w - w^*$, and write out the expression for the gradient in the limit of $\Delta w \to 0$ (see Appendix B for derivation)

$$
\begin{aligned}
\frac{\partial \mathcal{L}_1}{\partial w_1} \bigg|_{\Delta w \to 0} &= x^T \underbrace{(x \Delta w_1 \odot \rho'(xw_1) \odot \rho'(xw_1))}_{G_1} \\
\frac{\partial \mathcal{L}_2}{\partial w_1} \bigg|_{\Delta w \to 0} &= \underbrace{x^T \left( x \Delta w_1 \odot \rho'(xw_1) w_2 \odot \rho'(s_1 w_2)^2 w_2^T \odot \rho'(xw_1) \right)}_{G_2} \\
&\quad x^T \left( s_1 \Delta w_2 \odot \rho'(s_1 w_2)^2 w_2^T \odot \rho'(xw_1) \right)
\end{aligned} \tag{12}
$$

When the term $w_2 \odot \rho'(s_1 \cdot w_2)^2 \cdot w_2^T$ is proportional to an identity matrix, we can see that $\frac{\partial \mathcal{L}_1}{\partial w_1}$ and $G_1$ are perfectly aligned. This will be the case when $w_2$ is orthogonal and layer 2 has a linear activation. However, even for randomly sampled parameters and a nonlinear activation, $w_2 \odot \rho'(s_1 \cdot w_2)^2 \cdot w_2^T$ tends to have a strong diagonal component and the terms thus tend to be positively aligned. Figure 1 demonstrates that this gradient-alignment tends to *increase* as then network trains to approximate a set of targets (i.e. as $\phi \to \phi^*$). Note that the alignment of the *local* loss-gradient with the *global* loss-gradient is at least as high as with the *distant* loss-gradient, because $\nabla_\phi \mathcal{L}_{global} = \nabla_\phi \mathcal{L}_{local} + \nabla_\phi \mathcal{L}_{distant}$ and $\mathcal{S}(\nabla_\phi \mathcal{L}_{distant}, \nabla_\phi \mathcal{L}_{local}) \le S(\nabla_\phi \mathcal{L}_{distant} + \nabla_\phi \mathcal{L}_{local}, \nabla_\phi \mathcal{L}_{local}) \quad \forall \nabla_\phi \mathcal{L}_{local}, \nabla_\phi \mathcal{L}_{distant}$.

This explains the empirical observation in Figures 1 and 3 that optimizing the local, as opposed to the global, loss for the feedforward network does not appear to slow down convergence: Later layers do not have to "wait" for earlier layers to converge before they themselves converge - earlier layers

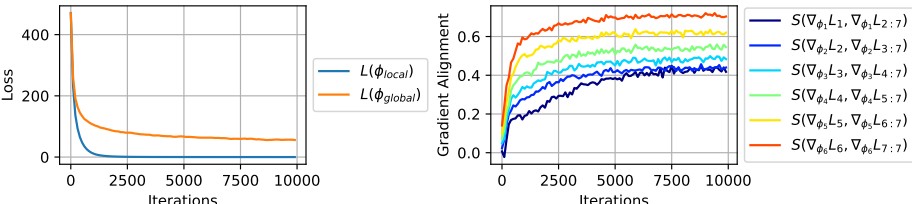

Figure 1: We train a 6-layer network with parameters $\phi$ to predict layerwise targets generated by another network with random parameters $\phi^*$. **Left:** We compare the convergence of the global loss of two training runs starting from the same initial conditions and identical (untuned) hyperparamters: A network with parameters $\phi_{local}$ trained using only local losses and a network with parameters $\phi_{global}$ trained directly on the global loss. We note that the *locally trained* network converges significantly faster, suggesting that optimization is easier in the absence of the "confusing" distant-gradient signals from the not-yet-converged higher layers. **Right:** We plot the cosine-similarity of local and distant components of the gradient of $\phi_{local}$ as training progresses. We see that as we approach convergence (as $\phi_{local} \to \phi^*$), the local and distant gradients tend to align.

optimize the loss of later layers right from the beginning of training. As shown in Figure 1, it may in fact speed up convergence since each layer's optimizer is solving a simpler problem (albeit with changing input representations for layers $> 1$).

When local targets $s^-$ are provided by the equilibrating network, it is not in general true that there exists some $\phi^*$ such that $s^- = s^*$. In our experiments, we observed that this did not prevent the forward network from learning to classify just as well as the equilibrating network. However, this may not hold for more complex datasets. As mentioned in Section 2.3, this could be resolved in future work with a scheme for annealing $\lambda$ up to infinity while maintaining stable training.

## 3 EXPERIMENTS

We base our experiments off of those of Scellier & Bengio (2017): We use the hard sigmoid $\rho(x) = max(0, min(1, x))$ as our nonlinearity. We clip the state of $s_i$ to the range (0, 1) because, since $\rho'(x) = 0 : x < 0 \lor x > 1$, if the system in Equation 2 were run in continuous time, it should never reach states outside this range. Borrowing a trick from Scellier & Bengio (2017) to avoid instability problems arising from incomplete negative-phase convergence, we randomly sample $\beta \sim \mathcal{U}(\{-\beta_{base}, +\beta_{base}\})$, where $\beta_{base}$ is a small positive number, for each minibatch and use this for both the positive phase and for multiplying the learning rate in Equation 3 (for simplicity, this is not shown in Algorithm 1). [3] . Unlike Scellier & Bengio (2017), we do not use the trick of caching and reusing converged states for each data point between epochs. In order to avoid "dead gradient" zones, we modify the activation function of our feedforward network (described in Equation 6) to $\rho^{mod}(x) = \rho(x) + 0.01x$, where the 0.01 "leak" is added to prevent the feed-forward neurons from getting stuck due to zero-gradients in the saturated regions. We use $\lambda = 0.1$ as the regularizing parameter from Equation 10, having scanned for values in Appendix D.

### 3.1 MNIST

We verify that the our algorithm works on the MNIST dataset. The learning curves can be seen in Figure 2. We find, somewhat surprisingly, that the forward pass of our network performs almost indistinguishably from the performance of the negative-phase of Equilibrium Propagation. This encouraging result shows that this approach for training a feedforward network without backprop does indeed work. We also see from from the top-two panels of Figure 2 that our approach can stabilize Equilibrium-Prop learning when we run the network for fewer steps than are needed for

---

[3]When $\beta$ is negative, the positive-state $s^+$ is pushed *away* from the targets, but gradients still point in the correct direction because the learning rate is scaled by $-1/\beta$. This trick avoids an instability when, due to incomplete negative-phase convergence, the network continues approaching the true minimum of $E(s, x)$ in the positive phase, and thus on every iteration contues to push *down* the energy of this "true" negative minimum

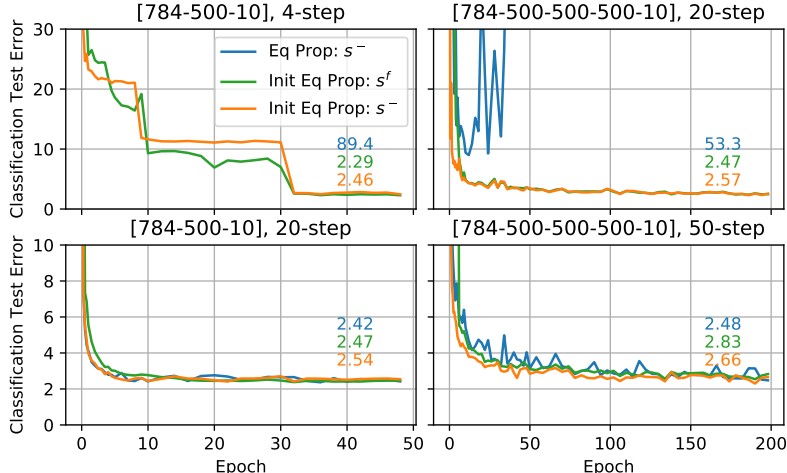

Figure 2: Learning Curves on MNIST comparing the performance of Equilibrium Propagation (Eq Prop: $s^-$), the Forward-Pass in Initialized Equilibrium Propagation (Fwd Eq Prop: $s^f$) (Algorithm 2) and the Negative Phase in Initialized Equilibrium Propagation (Fwd Eq Prop: $s^-$) (Algorithm 3) Numbers indicate error at the final test. **Left Column:** A shallow network with a single hidden layer of 500 units. **Right Column:** A deeper network with 3 layers of [500, 500, 500] hidden units. **Top Row:** Training with a small-number of negative-phase steps (4 for the shallow network, 20 for the deeper) shows that feedfoward initialization makes training more stable by providing a good starting point for the negative phase optimization. The *Eq Prop $s^-$* lines on the upper plots are shortened because we terminate training when the network fails to converge. **Bottom Row:** Training with more negative-phase steps shows that when the baseline Equilibrium Propagation network is given sufficient time to converge, it performs comparably with our feedforward network (Note that the y-axis scale differs from the top).

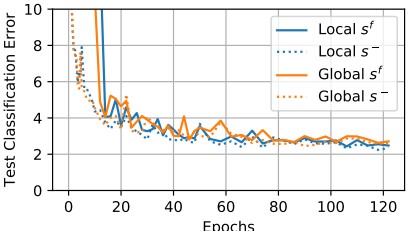
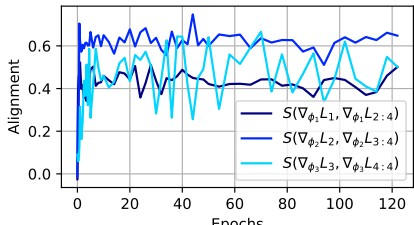

Figure 3: Test scores and gradient alignment on [784-500-500-500-10] network trained on MNIST **Left:** We compare the performance of Initialized Equilibrium Propagation when the feedforward network is trained using only local losses vs the global loss (i.e. using backpropagation). $s^f$ denotes the forward pass and $s^-$ denotes the state at the end of the negative phase. Note that we observe no disadvantage when we only use local losses. **Right:** We observe the same effect as for our toy problem (see Figure 1). Early on in training, the local error gradients tend to align with gradients coming from higher layers.

full convergence. By initializing the negative phase in a close-to-optimal regime, the network is able to learn when the number of steps is so low that plain Equilibrium Propagation cannot converge. Moreover we note that as the number of steps is enough for convergence, there is not much advantage to using more negative-phase iterations - the longer negative phase does not improve our error.

In Figure 3 we demonstrate that using only local losses to update the feedforward network comes with no apparent disadvantage. In line with our results from Section 2.4, we see that local loss gradients become aligned with the loss gradients from higher layers, explaining why it appears to be sufficient to only use the local gradients.

## 4 RELATED WORK

The most closely related work to ours is by Bengio et al. (2016). There, the authors examine the idea of initializing an iterative settling process with a forward pass. They propose using the parameters of the Equilibriating network to do a forward pass, and describe the conditions under which this provides a good approximation of the energy-minimizing state. Their conclusion is that this criterion is met when consecutive layers of the energy-based model form a good autoencoder. Their model differs from ours in that the parameters of the forward model are tied to the parameters of the energy-based model. The effects of this assumption are unclear, and the authors do not demonstrate a training algorithm using this idea.

Our work was loosely inspired by Hinton et al. (2015), who proposed "distilling" the knowledge of a large neural network or ensemble into a smaller network which is designed to run efficiently at inference time. In this work, we distill knowledge from a slow, equilibrating network in to a fast feedforward network.

### 4.1 RELATION TO ADVERSARIAL LEARNING

Several authors (Kim & Bengio, 2016), (Finn et al., 2016), (Zhai et al., 2016) have pointed out the connection between Energy Based Models and Generative Adversarial Networks (GANs). In these works, a feedforward *generator* network proposes synthetic samples to be evaluated by an energy-based *discriminator*, which learns to push down the energy of real samples and push up the energy of synthetic ones. In these models, both the *generator/sample proposer* and the *discriminator/energy-based-model* are deep feedforward networks trained with backpropagation.

In our approach, we have a similar scenario. The inference network $f_\phi$ can be thought of as a conditional generator which produces a network state $s^f$ given a randomly sampled input datum $x$: $s^f = f_\phi(x)$. Parameters $\phi$ are trained to approximate the minimal-energy states of the energy function: $\min_\phi \| f_\phi(x) - \arg\min_s E_\theta(s, x) \|$. However, in our model, the Energy-Based network $E_\theta(s, x)$ does not directly evaluate the energy of the generated data $s^f$, but of the minimizing state $s^- = \arg\min_s E_\theta(s, x)$ which is produced by performing $T^-$ energy-minimization steps on $s^f$ (see Algorithm 1). Like a discriminator, the energy-based model parameters $\theta$ learn based on a contrastive loss which pushes up the energy of the "synthetic" network state $s^-$ while pushing down the energy of the "real" state $s^+$.

### 4.2 RELATION TO AMORTIZED VARIATIONAL INFERENCE

In variational inference, we aim to estimate a posterior distribution $p(z|x)$ over a latent variable $z$ given data $x$, using an *approximate posterior* $q(z)$. Algorithms such as Expectation Maximization (Dempster et al., 1977) iteratively update a set of posterior parameters $\mu$ per-data point, so that $z_n \sim q(z|\mu_n)$. In *amortized inference*, we instead learn a global set of parameters $\phi$ which can map a sample $x_n$ to a posterior estimate $z_n \sim q_\phi(z|x_n)$. Dayan et al. (1995) proposed using a "recognition' network" as this amortized predictor, and Kingma & Welling (2013) showed that you can train this recognition network efficiently using the reparameterization trick. However, this comes at the expense of an "amortization gap" (Cremer et al., 2018) - where the posterior estimate suffers due to the sharing of posterior estimation parameters across data samples. Several recent works (Marino et al., 2018), (Li et al., 2017), (Kim et al., 2018), have proposed various versions of a "teacher-student" framework, in which an amortized network $q_\theta(z|x)$ provides an initial guess for the posterior, which is then refined by a slow, non-amortized network which refines $q(z)$ in several steps into a better posterior estimate. The "student" then learns to refine its posterior estimate using the final result of the iterative inference. In the context of training Deep Boltzmann Machines, Salakhutdinov & Larochelle (2010) trained a feedforward network with backpropagation to initialize variational parameters which are then optimized to estimate the posterior over latent variables.

Initialized Equilibrium Propagation is a zero-temperature analog of amortized variational inference. In the zero-temperature limit, the mean-field updates of variational inference reduce to coordinate ascent on the variational parameters. The function of the amortized student network $q_\phi(z|x)$ is then analogous to the function of our initializing network $f_\phi(x)$, and the negative phase corresponds to the iterative optimization of varational parameters from the starting point provided by $f_\phi(x)$.

### 4.3 RELATION TO OTHER WORK IN LOCAL CREDIT-ASSIGNMENT

Another interesting approach to shortening the inference phase in Equilibrium propagation was proposed by Kohan et al. (2018). The authors propose a model that is *almost* a feedforward network, except that the output layer projects back to the input layer. The negative phase consists of making several feedforward passes through the network, reprojecting the output back to the input with each pass. Although the resulting inference model is not a feedforward network, the authors claim that this approach allows them to dramatically shorten convergence time of the negative phase.

There is also a notable similarity between Initialized Equilibrium Propagation and Method of Auxiliary Coordinates (Carreira-Perpinan & Wang, 2014). In that paper, the authors propose a scheme for optimizing a layered feedforward network which consists of alternating optimization of the neural activations (which can be parallelized across samples) and parameters (which can be parallelized across layers). In order to ensure that the layer activations $z_k$ remain close to what a feedforward network can compute, the objective includes a layerwise cost $\frac{\mu}{2}\|z_k - f_k(z_{k-1})\|^2$, where $z_k$ is layer $k$'s activation, $f_k$ is layer $k$'s function, and $\mu$ is a the strength of the layerwise cost (as they anneal $\mu \to \infty$ this cost becomes a constraint). This is identical in form and function to our $\lambda \sum_{j \in \mathcal{S}} (s^f - s^-)^2$ term in Equation 10. However, they differ from our method in that their neurons backpropagate their gradients back to input neurons (albeit only across one layer). Taylor et al. (2016) do something similar with using the Alternating Direction Method of Multipliers (ADMM), where Lagrange multipliers enforce the "layer matching" constraints exactly. Both methods, unlike Equilibrium Prop, are full-batch methods.

More broadly, other approaches to backprop-free credit assignment have been tried. Difference-Target propagation (Lee et al., 2015) proposes a mechanism to send back targets to each layer, such that locally optimizing targets also optimizes the true objective. Feedback-Alignment (Lillicrap et al., 2014) shows that, surprisingly, it is possible to train while using random weights for the backwards pass in backpropagation, because the forward pass parameters tends to "align" to the backwards-pass parameters so that the pseudogradients tend to be within $90°$ of the true gradients. A similar phenomenon was observed in Equilibrium Propagation by Scellier et al. (2018), who showed that when one removed the constraint of symmetric weight in Equilibrium propagation, the weights would evolve towards symmetry through training. Finally, Jaderberg et al. (2016) used a very different approach - rather than create local targets, each layer predicts its own "pseudogradient". The gradient prediction parameters are then trained either by the true gradients (which no longer need to arrive before a parameter update takes place) or by backpropagated versions of pseudogradients from higher layers.

## 5 DISCUSSION

In this paper we describe how to use a recurrent, energy-based model to provide layerwise targets with which to train a feedforward network without backpropagation. This work helps us understand how the brain might be training fast inference networks. In this view, neurons in the inference network learn to predict local targets, which correspond to the minimal energy states, which are found by the iterative settling of a separate, recurrently connected *equilibrating network*.

More immediately perhaps, this could lead towards efficient analog neural network designs in hardware. As pointed out by Scellier & Bengio (2017), it is much easier to design an analog circuit to minimize some (possibly unknown) energy function than it is to design a feedforward circuit and a parallel backwards circuit which exactly computes its gradients. However it is very undesirable for the function of a network to depend on peculiarities of a particular piece of analog hardware, because then the network cannot be easily replicated. We could imagine using a hybrid circuit to train a digital, copy-able feedforward network, which is updated by gradients computed in the analog hardware. Without the constraint of having to backpropagate through the feedforward network, designs could be simplified, for example to do away with the need for differentiable activation functions or to use feedforward architectures which would otherwise suffer from vanishing/exploding gradient effects.

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

# A  GLOSSARY

Here we we have a reference of symbols used in the paper, in (Greek, Latin) alphabetical order.

$\alpha_i \subset \{j : j \in \mathcal{S}, j \neq i\}$: The set of neurons in the Equilibrating Network that connect to neuron $i$

$\alpha_i^f = \{j : j \in \alpha_i, j < i\}$: The set of neurons in the Feedforward Network connected to neuron $i$.

$\beta \in \mathbb{R}$: The perturbation factor, which modulates how much the *augmented energy* $E_\theta^\beta$ is affected by the output loss.

$\eta \in \mathbb{R}^+$: The learning rate.

$\theta$: The set of parameters (all $w_{ij}$'s and $b_j$'s, in the Equilibrating network)

$\rho$: a neuron nonlinearity. In all experiments it is $\rho(x) = max(0, min(1, x))$

$\phi = (\omega, c)$: The set of parameters (all $\omega_{ij}$'s and $c_j$'s), in the feedforward network

$\phi_j$: The set of parameters (all $\omega_{\cdot j}$'s and $c_j$') belonging to a neuron $j$

$(\omega, c)$: The (weights, biases) of the feedforward network. Collectively called $\phi$

$C(s_\mathcal{O}, y) \in \mathbb{R}$: The output loss function, defined on the states of the output units.

$E_\theta(s, x) \in \mathbb{R}$: The energy function of the Equilibrating network (Equation 1) produces a scalar energy given a set of states $s$ and input $x$

$E_\theta^\beta(s, x, y) = E_\theta(s, x) + \beta \frac{\partial C(s_\mathcal{O}, y)}{\partial s} \in \mathbb{R}$: The *augmented* energy function of the Equilibrating network, when it has been perturbed by a factor $\beta$ by target data $y$

$f_\phi(x) \mapsto s^f$: The *initialization* function: A feedforward network which initializes the state of the Equilibrating network.

$\mathcal{I}$: The set of indices of input neurons.

$\mathcal{L}$: The total loss in the Feedforward network's prediction. Defined in Equation 8

$\mathcal{L}_i$: The local loss on the $i'th$ neuron in the feedforward network. Defined in Equation 8

$\mathcal{O}$: The set of indices output neurons, a subset of $\mathcal{S}$

$\mathcal{S}$: The set of inidices non-input neurons.

$s$: The set of neuron states. $s := \{s_i : i \in \mathcal{S}\} \in \mathbb{R}^{\dim(\mathcal{S})}$

$s^- := \arg\min_s E(s, x) \in \mathbb{R}^{\dim(\mathcal{S})}$: The minimizing state of the Energy function.

$s^+ := \arg\min_s E_\theta^\beta(s, x, y) \in \mathbb{R}^{\dim(\mathcal{S})}$ The minimizing state of the augmented energy function.

$s^f := f_\phi(x)z \in \mathbb{R}^{\dim(\mathcal{S})}$: The state output by the feedforward network.

$s_\mathcal{O} \in \mathbb{R}^{\dim(\mathcal{O})}$: the states of the output units

$T^-, T^+$: Hyperparameters for Equilibrium Prop defining the number of steps of convergence of the negative/positive phase.

$w, b$: the parameters of the Equilibrating network (collectively called $\theta$)

$x \in \mathbb{R}^{\dim \mathcal{I}}$: The input data

$y \in \mathbb{R}^{\dim \mathcal{O}}$: The target data

## B    GRADIENT ALIGNMENT

Here to derive the result in Equation 12. First, we restate Equation 11 substituting $w^* = w + \Delta w$:

$$s_1 = \rho(xw_1) \qquad s_1^* = \rho(x(w_1 - \Delta w_1)) \qquad \mathcal{L}_1 = \|s_1 - s_1^*\|_2^2$$
$$s_2 = \rho(s_1 w_2) \qquad s_2^* = \rho(s_1^*(w_2 - \Delta w_2)) \qquad \mathcal{L}_2 = \|s_2 - s_2^*\|_2^2$$

Where:

$$x \in \mathbb{R}^{N \times D_0}; \, s_1, s_1^* \in \mathbb{R}^{N \times D_1}; \, s_2, s_2^* \in \mathbb{R}^{N \times D_2}; \, w_1, \Delta w_1 \in \mathbb{R}^{D_0 \times D_1}; \, w_2, \Delta w_2 \in \mathbb{R}^{D_1 \times D_2}$$

Now we will compute the gradient of each of the local losses with respect to $\Delta w_1$, in the limit where $\Delta w_1$ is small.

$$\frac{\partial \mathcal{L}_1}{\partial w_1} = \frac{\partial \mathcal{L}_1}{\partial s_1}\frac{\partial s_1}{\partial w_1}$$

$$= \left( \left( (s_1 - s_1^*) \odot \rho'(xw) \right)^T \cdot x \right)^T$$

$$= x^T \overbrace{\left( (\rho(xw_1) - \rho(x(w_1 - \Delta w_1))) \odot \rho'(xw_1) \right)}^{g(\Delta w_1)}$$

$$\overset{\lim_{\Delta w \to 0}}{=} g(0) + \Delta w_1 \frac{\partial g}{\partial \Delta w_1}(0) \qquad \text{(1st order Taylor Expansion about } \Delta w_1 = 0)$$

$$= x^T \left( \rho(x(w_1 - 0)) - \rho(xw_1))\rho'(xw_1) \right) + x^T \left( x\Delta w_1 \odot \rho'(xw_1)\rho'(xw_1) \right)$$

$$= 0 + x^T \left( x\Delta w_1 \odot \rho'(xw_1)^2 \right)$$

$$\frac{\partial \mathcal{L}_2}{\partial w_1} = \frac{\partial \mathcal{L}_2}{\partial s_2}\frac{\partial s_2}{\partial s_1}\frac{\partial s_1}{\partial w_1}$$

$$= \left( \left( (s_2 - s_2^*) \odot \rho'(s_1 w_2)w_2^T \rho'(xw_1) \right)^T x \right)^T$$

$$= x^T \left( (s_2 - s_2^*) \odot \rho'(s_1 w_2)w_2^T \rho'(xw_1) \right)$$

$$= x^T \overbrace{\left( \left( \rho(\rho(xw_1)w_2) - \rho(\rho(x(w_1 - \Delta w_1))(w_2 - \Delta_{w_2})) \right) \odot \rho'(\rho(xw_1)w_2)w_2^T \rho'(xw_1) \right)}^{g(\Delta w_1, \Delta w_2)}$$

$$\overset{\lim_{\Delta w \to 0}}{=} g(0,0) + \Delta w_1 \frac{\partial g}{\partial \Delta w_1}(0,0) + \Delta w_2 \frac{\partial g}{\partial \Delta w_2}(0,0) \qquad \text{(1st order Taylor Expansion about } \Delta w_1 = 0, \Delta w_2 = 0)$$

$$= 0 + \overbrace{x^T \left( x\Delta w_1 \odot \rho'(xw_1)w_2 \odot \rho'(s_1 w_2)^2 w_2^T \odot \rho'(xw_1) \right)}^{G_1} + \overbrace{x^T \left( s_1 \Delta w_2 \odot \rho'(s_1 w_2)^2 w_2^T \odot \rho'(xw_1) \right)}^{G2}$$

## C   GRADIENT ALIGNMENT AT INITIALIZATION

Why do we observe gradient alignment even at random initialization? Let us start with the same 2-layer network defined in Appendix B

$$\frac{\partial \mathcal{L}_1}{\partial w_1} = \frac{\partial \mathcal{L}_1}{\partial s_1} \frac{\partial s_1}{\partial w_1}$$

$$= \left( \left( (s_1 - s_1^*) \odot \rho'(xw) \right)^T \cdot x \right)^T$$

$$= x^T \left( \left( \rho(xw_1) - \rho(xw_1^*) \right) \odot \rho'(xw_1) \right)$$

$$= \overbrace{x^T \rho(xw_1) \odot \rho'(xw_1)}^{G_A} - x^T \rho(xw_1^*) \odot \rho'(xw_1)$$

$$\frac{\partial \mathcal{L}_2}{\partial w_1} = \frac{\partial \mathcal{L}_2}{\partial s_2} \frac{\partial s_2}{\partial s_1} \frac{\partial s_1}{\partial w_1}$$

$$= \left( \left( (s_2 - s_2^*) \odot \rho'(s_1 w_2) w_2^T \rho'(xw_1) \right)^T x \right)^T$$

$$= x^T \left( (s_2 - s_2^*) \odot \rho'(s_1 w_2) w_2^T \rho'(xw_1) \right)$$

$$= x^T \left( \left( \rho(\rho(xw_1)w_2) - \rho(\rho(xw_1^*)w_2^*) \right) \odot \rho'(\rho(xw_1)w_2) w_2^T \rho'(xw_1) \right)$$

$$= \overbrace{x^T \left( \rho(\rho(xw_1)w_2) \odot \rho'(\rho(xw_1)w_2) w_2^T \rho'(xw_1) \right)}^{G_B} -$$

$$x^T \left( \rho(\rho(xw_1^*)w_2^*) \odot \rho'(\rho(xw_1)w_2) w_2^T \rho'(xw_1) \right)$$

$G_A$ and $G_B$ tend to be aligned because the terms $\rho(xw_1)$ and $\rho(\rho(xw_1)w_2) \odot \rho'(\rho(xw_1)w_2)w_2^T$ tend to be aligned. Suppose $\rho$ is a piecewise saturating nonlinearity (as we have in this paper) with $\rho(x) = [a \text{ if } (x < a); x \text{ if } (x \in [a, b]); b \text{ otherwise}]$

Then we can define a weight matrix $w_2'$ by filtering rows of $w_2$ to only include weights projecting to non-saturated neurons: $w_2' = [w_2^{(i)} \forall i : \rho'(\rho(xw_1)w_2^{(i)}) \neq 0]$ Where $w_2^{(i)}$ denotes the $i'th$ row of $w_2$.

Then our second term can be rewritten as: $\rho(\rho(xw_1)w_2) \odot \rho'(\rho(xw_1)w_2)w_2^T = \rho(xw_1)w_2'w_2'^T$. Given a random matrix $w_2'$, the matrix $w_2'w_2'^T$ will tend to have a strong diagonal component, causing this term to be aligned with $\rho(xw_1)$

# D  EFFECT OF $\lambda$ PARAMETER

In Equation 10 we introduce a new parameter $\lambda$ which encourages the state of the equilibrating network to state close to that of the forward pass. Here we perform a sweep of parameter $\lambda$ to evaluate its effect.

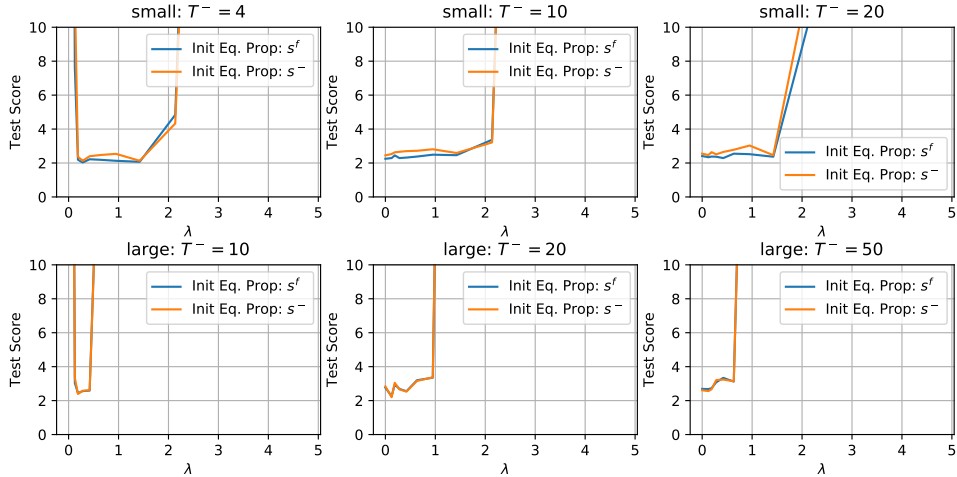

Figure 4: Here we scan the $\lambda$ parameter and plot the final score at the end of training. Each point in each plot corresponds to the final score of a network with parameter $\lambda$ fixed at the given value throughout training. The top row of plots is a for a small network with one hidden layer of 500 hidden units. The bottom is for a large network with 3 layers of [500, 500, 500] hidden units. Each column is for a different number of steps of negative-phase convergence.

We see in Figure 4 that when the number of steps of negative-phase convergence is small, introducing $\lambda$ can allow for more stable training. This makes sense - if the minimizing state of the equilibating network is "pulled" towards the state at the forward pass, it should take fewer steps of iteration to reach this state when initialized at the state of the forward pass. However, we also see that training fails when $\lambda$ is too high. We believe this is because the simple iterative settling scheme (Euler integration) used in this paper, as well as the original Equilibrium Prop by Scellier & Bengio (2017), can become unstable when optimizing a loss surface with sharp, steep, minima (as are induced with large $\lambda$). This could be addressed in later work by either using an adaptive $\lambda$ term or an adaptive Euler-integration step-size.

