# OpenReview forum: "Initialized Equilibrium Propagation for Backprop-Free Training"
_ICLR.cc/2019/Conference_

### Official Review · AnonReviewer1 · 2018-10-31

**Rating:** 7
**Confidence:** 5

**Review:**

This paper presents an improvement on the local/derivative-free learning algorithm equilibrium propagation. Specifically, it trains a feedforward network to initialize the iterative optimization process in equilibrium prop, leading to greater stability and computational efficiency, and providing a network that can later be used for fast feedforward predictions on test data. Non-local gradient terms are dropped when training the feedforward network, so that the entire system still doesn't require backprop. There is a neat theoretical result showing that, in the neighborhood of the optimum, the dropped non-local gradient terms will be correlated with the retained gradient terms.

My biggest concern with this paper is the lack of significant literature review, and that it is not placed in the context of previous work. There are only 12 references, 5 of which come from a single lab, and almost all of which are to extremely recent papers. Before acceptance, I would ask the authors to perform a literature search, update their paper to include citations to and discussion of previous work, and better motivate the novelty of their paper relative to previous work. Luckily, this is a concern that is addressable during the rebuttal process! If the authors perform a literature search, and update their paper appropriately, I will raise my score as high as 7.

Here are a few related topic areas which are currently not discussed in the paper. *I am including these as a starting point only! It is your job to do a careful literature search. I am completely sure there are obvious connections I'm missing, but these should provide some entry points into the citation web.*
- The "method of auxiliary coordinates" introduces soft (often quadratic) couplings between post- and pre- activations in adjacent layers which, like your distributed quadratic penalty, eliminate backprop across the couplings. I believe researchers have also done similar things with augmented Lagrangian methods. A similar layer-local quadratic penalty also appears in ladder networks.
- Positive/negative phase (clamped / unclamped phase) training is ubiquitous in energy based models. Note though that it isn't used in classical Hopfield networks. You might want to include references to other work in energy based models for both this and other reasons. e.g., there may be some similarities between this approach and continuous-valued Boltzmann machines?
- In addition to feedback alignment, there are other approaches to training deep neural networks without standard backprop. examples include: synthetic gradients, meta-learned local update rules, direct feedback alignment, deep Boltzmann machines, ...
- There is extensive literature on biologically plausible learning rules -- it is a field of study in its own right. As the paper is motivated in terms of biological plausibility, it would be good to include more general context on the different approaches taken to biological plausibility.

More detailed comments follow:

Thank you for including the glossary of symbols!

"Continuous Hopfield Network" use lowercase for this (unless introducing acronym)

"is the set non-input" -> "is the set of non-input"

"$\alpha = ...$ ... $\alpha_j \subset ...$" I could not make sense of the set notation here.

would recommend using something other than rho for nonlinearity. rho is rarely used as a function, so the prior of many readers will be to interpret this as a scalar. phi( ) or f( ) or h( ) are often used as NN nonlinearities.

inline equation after "clamping factor" -- believe this should just be C, rather than \partial C / \partial s.
Move definition of \mathcal O up to where the symbol is first used.

text before eq. 7 -- why train to approximate s- rather than s+? It seems like s+ would lead to higher accuracy when this is eventually used for inference.

eq. 10 -- doesn't the regularization term also decrease the expressivity of the Hopfield network? e.g. it can no longer engage in "explaining away" or enforce top-down consistency, both of which are powerful positive attributes of iterative estimation procedures.

notation nit: it's confusing to use a dot to indicate matrix multiplication. It is commonly used in ML to indicate an inner product between two vectors of the same shape/orientation. Typically matrix multiplication is implied whenever an operator isn't specified (eg x w_1 is matrix multiplication).

eq. 12 -- is f' supposed to be h'? And wasn't the nonlinearity earlier introduced as rho? Should settle on one symbol for the nonlinearity.

This result is very cool. It only holds in the neighborhood of the optimum though. At initialization, I believe the expected correlation is zero by symmetry arguments (eg, d L_2 / d s_2 is equally likely to have either sign). Should include an explicit discussion of when this relationship is expected to hold.

"proportional to" -> "correlated with" (it's not proportional to)

sec. 3 -- describe nonlinearity as "hard sigmoid"

beta is drawn from uniform distribution including negative numbers? beta was earlier defined to be positive only.

Figure 2 -- how does the final achieved test error change with the number of negative-phase steps? ie, is the final classification test error better even for init eq prop in the bottom row than it is in the top?

The idea of initializing an iterative settling process with a forward pass goes back much farther than this. A couple contexts being deep Boltzmann machines, and the use of variational inference to initialize Monte Carlo chains

sect 4.3 -- "the the" -> "to the"

---

> ### Author Response · Authors · 2018-11-26
> **Response**
>
> Thank you for your flexible review system which incentivized us to do a proper exploration of related work!  In response to your concern, we have significantly expanded the related work section.  We now mention what we think is an important link to amortized variational inference.  We have also simplified notation in response to your other comments.  Below we respond to some of your more detailed comments:
>
> ---
> - “would recommend using something other than rho for nonlinearity. rho is rarely used as a function, so the prior of many readers will be to interpret this as a scalar. phi( ) or f( ) or h( ) are often used as NN nonlinearities.”
>
> In Equilibrium Propagation and other related papers rho is used as the nonlinearity.  So we would like to keep rho here in order to be consistent with those papers.
> ----
> - “why train to approximate s- rather than s+? It seems like s+ would lead to higher accuracy when this is eventually used for inference.”
>
> Good observation - we added a footnote to address it: We could also minimize the distance with s+, but found experimentally that this actually works slightly worse than s−. We believe that this is because equilibrium propagation depends on the s− being very close to a true minimum of the energy function, and so initializing the negative phase to sf≈s− will lead to better gradient computations than when we initialize the negative phase to sf≈s+
>
> ----
>
> - “eq. 10 -- doesn't the regularization term also decrease the expressivity of the Hopfield network? e.g. it can no longer engage in "explaining away" or enforce top-down consistency, both of which are powerful positive attributes of iterative estimation procedures.”
>
> It does potentially reduce the expressivity, for the reasons you describe.  We now expand a bit about that in Section 2.3.  However, our primary concern here is to train a feedforward network without backpropagation, so we already accept that we’re producing a model with the expressivity of a feedforward network.  That said, because of concerns about this, we add an experiment in Appendix C where we run training to completion for various swept values of lambda.  We do observe that too-high a lambda causes eq.prop to become unstable and fail, but we do not observe what we might expect if constrained-expressivity were a problem: (i.e. the error for s^f decreasing and the error for s- increasing with increased lambda).
>
> ----
>
> - “This result is very cool. It only holds in the neighborhood of the optimum though. At initialization, I believe the expected correlation is zero by symmetry arguments (eg, d L_2 / d s_2 is equally likely to have either sign). Should include an explicit discussion of when this relationship is expected to hold.”
>
> Thank you!  We also found it unintuitive at first.  Even at random initialization the gradient alignment S(d_L1/d_phi1, d_L2/d_phi1) is in general NOT zero-mean.  For a randomly initialized network, d_L2/d_phi1 and d_L1/d_phi1 are both zero-mean random variables (for the reason you mention - that d_L2/d_s2 is equally likely to have either sign), but they are not independent - they are both functions of phi1, and this dependency induces alignment.  We’ve changed Figure to demonstrate that the initially weak alignment becomes stronger as training progresses (and phi approaches phi*), and added a derivation of the alignment result in Appendix B.
>
> ----
> - "proportional to" -> "correlated with" (it's not proportional to)
>
> Our statement is “when the term is proportional to an identity matrix, we see that dL1/dw1 and G1 are perfectly aligned”.  This is true: we’re just describing the case in which ideal alignment happens, not saying that in a normal situation it’s proportional.
> . ---
>
> - “Figure 2 -- how does the final achieved test error change with the number of negative-phase steps? ie, is the final classification test error better even for init eq prop in the bottom row than it is in the top?”
>
> We modified Figure 2 to show the scores.  Answer is no, once there are enough negative steps for training to be stable, the forward pass doesn’t become more accurate with additional steps.

---

> > ### Comment · AnonReviewer1 · 2018-12-05
> > **paper greatly improved, some remaining questions/concerns**
> >
> > Thank you for the updates! The paper is much improved. I have raised my score. I still have some specific concerns, below:
> >
> > In the new Figure 1a, could you talk about how you search over optimization and initialization hyper-parameters for the local and global loss cases? I have a suspicion that the better performance of the local network may be due only to hyperparameters being better tuned for local rather than global training.
> >
> > re "they differ from our method in that their layer activations z_k are calculated using backpropagation-based optimization."
> > I'm pretty sure this is not an accurate statement about the method of auxiliary coordinates? At least, it is my understanding that, due to the quadratic coupling between z_k in adjacent layers, the gradient with respect to z_k only depends on the z_{k-1 ... k+1}, and so there is no backpropagation through multiple layers in the network. Similarly for the gradient with respect to the weights in a given layer.
> >
> > More minor questions/comments:
> >
> > re: "Even at random initialization the gradient alignment S(d_L1/d_phi1, d_L2/d_phi1) is in general NOT zero-mean.  For a randomly initialized network, d_L2/d_phi1 and d_L1/d_phi1 are both zero-mean random variables (for the reason you mention - that d_L2/d_s2 is equally likely to have either sign), but they are not independent - they are both functions of phi1, and this dependency induces alignment.  We’ve changed Figure to demonstrate that the initially weak alignment becomes stronger as training progresses (and phi approaches phi*), and added a derivation of the alignment result in Appendix B."
> >
> > Can you say more about why you would expect the gradients to be aligned at initialization? Is it just that both gradients are expected to have a non-zero projection in the phi1 direction (because the distance between two random vectors will tend to shrink if either vector is moved towards the origin)?
> >
> > Another thread of references that comes to mind, for biologically plausible local learning rules in machine learning, is the use of meta-learning to learn those rules. A seed paper is:
> > Yoshua Bengio, Samy Bengio, and Jocelyn Cloutier. Learning a synaptic learning rule. Université de Montréal,
> > Département d’informatique et de recherche opérationnelle, 1990.

---

> > > ### Author Response · Authors · 2018-12-05
> > > **Response**
> > >
> > > - "In the new Figure 1a, " ... "I have a suspicion that the better performance of the local network may be due only to hyperparameters being better tuned for local rather than global training."
> > >
> > > No hyperparameter search was done for this experiment.  The only hyperparameter that we tuned was the scale of the initial weights / target-network weights, and that was tuned to ensure approximately equal activation magnitudes for each layer (so that activations neither die out or saturate with depth).  The network had 8 layers (including input) of 200 units, training was done with SGD with learning rate of 0.01 and momentum of 0.9, but the results look the same with other optimizers we tried (Adagrad, Adamax). When we used no adaptation in the step size (ie momentumless SGD), the local started off slower (presumably because the gradients were simply smaller), but then overtook the global optimiser.
> > >
> > > - "I'm pretty sure this is not an accurate statement about the method of auxiliary coordinates?"
> > >
> > > You are correct, I'd misread the Z-step of MAC to be sample-local but layer-global, when in fact it is both sample-local and (sort of) layer local (still involves backprop across one layer).  We will update the draft for the final.  Thank you for pointing that out.
> > >
> > > - "Can you say more about why you would expect the gradients to be aligned at initialization?"
> > >
> > > It comes down to the fact that for a random matrix w, <x, x w w^T> will tend to be positive.  At random initialization, a component of d_L1/d_w1 and a component of  d_L2/d_w1 are related by a (w w^T) term, so tend to be positively aligned.  We have posted an (anonymous) explanation here, and will add it to the appendix: https://srv-file1.gofile.io/download/VfLgmv/2ceb9fae950fd7ab1b532534f4cd2435/Why-Alignment-at-Init.pdf
> > >
> > > In any case, the initial local-distant alignment is not that important, because even if it were random, the local gradient will still be aligned with the *global* gradient (which is the local+distant gradient) initially.  So from the start, phi is being pushed towards phi*, which in turn leads to an increased local-distant alignment.
> > >
> > > - "Another thread of references that comes to mind, for biologically plausible local learning rules"...
> > >
> > > Thank you for the paper.  We read this one, as well as the more recent "Learning Unsupervised Learning Rules" and a few other "Learning to Learn" papers. We don't (yet) see a close connection to the method in this work - both still require backpropagation (or a less efficient non-gradient-based search method) to train the parameters of the optimizer.  However we agree that this deserves mention as another line of work that may lead to biologically plausible local learning rules.

---

> > > > ### Comment · AnonReviewer1 · 2018-12-05
> > > > **weaken figure 1Left statement?**
> > > >
> > > > Thank you for your additional responses, and the added argument motivating gradient alignment.
> > > >
> > > > In terms of Figure 1, I would recommend modifying
> > > > "Perhaps surprisingly, the locally trained model converges faster. This is likely because
> > > > learning to optimize local targets is a simpler problem."
> > > > and noting that no hyper-parameter tuning has been performed, and that the relative performance of local vs. global learning thus remains an open question, with the current comparison at best suggestive.
> > > >
> > > > Otherwise -- I am out of comments. Interesting paper.  :)

---

> > > > > ### Author Response · Authors · 2018-12-06
> > > > > **Will do**
> > > > >
> > > > > We will use your suggestion to update the final draft, and in general, we'll do a proof-read over the paper to make sure we're not making overly-confidant claims.  Thank you for your constructive feedback throughout this review process.

---

> > > > > > ### Comment · AnonReviewer1 · 2018-12-19
> > > > > > **final score update**
> > > > > >
> > > > > > Have bumped score to 7, in anticipation of the final improvements from this thread being included in the camera ready.

---

### Official Review · AnonReviewer3 · 2018-11-01
**Init EqProp**

**Rating:** 8
**Confidence:** 5

**Review:**

This is a nice improvement on Equilibrium Propagation (EqProp) based on training a separate network to initialize (and speed-up at test time) the recurrent network trained by EqProp. The feedforward network takes as laywerwise targets the activities of each layer when running the recurrent net to convergence (s-). The surprising result (on MNIST) is that the feedforward approximation does as well as the recurrent net that trains it. This allows faster run-time, which is practically very useful.

My main concern is with the mathematical argument in section 2.2. s* is not the same as s- , and in general, it is not clear at all that there should be a phi* such that s*=s-. Also, the derivation in eqn 12 assumes that w is very close to w*, which is not clear at all. So this derivation is more suggestive, and the empirical results are the ones which could be convincing. My only concern there is that the only experiments performed are on MNIST, which is known to be easily dealt with using the kind of feedforward architectures studied here. Things could break down if much more non-linearity (which is what the fixed point recurrence provides) is necessary (equivalently this would correspond to networks for which much more depth is necessary, given some budget of number of parameters). I don't think that this is a deal-breaker, but I think that this section needs to be more prudent in the way that it concludes from these observations (the math and the experiments).

One question I have is about biological plausibility. The whole point of EqProp was to produce a biologically plausible variation on backprop. How plausible is it to have two sets of weights for the feedforward and recurrent parts? That is where a trick such as proposed in Bengio et al 2016 might be useful, so that the same set of weights could be used for both.

It might be good to mention Bengio et al 2016 in the introduction since it is the closest paper (trying to solve the same problem of using a feedforward net to approximate the true recurrent computation), rather than pushing that to the end.

In sec. 1.1, I would replace 'training a Continuous Hopfield Network for classification' by 'energy-based models, with a recurrent net's updates corresponding to gradient descent in the energy'. The EqProp algorithm is not just for the Hopfield energy but is general. Then before eq 1, mention that this is the variant of Hopfield energy studied in the EqProp paper.

I found a couple of typos (scenerio, of the of the).

---

> ### Author Response · Authors · 2018-11-26
> **Response**
>
> Thank you for your helpful review.  We agree with all the points you made, and have addressed them in the paper.  Below we address some of your questions individually:
> ----
>
> - “My main concern is with the mathematical argument in section 2.2. s* is not the same as s- , and in general, it is not clear at all that there should be a phi* such that s*=s-. Also, the derivation in eqn 12 assumes that w is very close to w*, which is not clear at al” …  I don't think that this is a deal-breaker, but I think that this section needs to be more prudent in the way that it concludes from these observations (the math and the experiments).
>
> This is definitely a valid concern.  We’ve added a paragraph to the end of this section (now 2.4) addressing your point.  We also mention that in future work, this problem could be dealt-with by figuring out a smart way to anneal the lambda-term (Introduced in now-section 2.3: Including the forward states in the energy function) in a way that does not harm training of the equilibrating network.  We note that in the limit of lambda -> infinity, s-=sf, and so the targets provided by the equilibrating network are achievable by the forward network.  However we observe experimentally (see added experiment in Appendix C) that setting lambda too high can cause training instabilities.
>
> -----
>
> - “One question I have is about biological plausibility. The whole point of EqProp was to produce a biologically plausible variation on backprop. How plausible is it to have two sets of weights for the feedforward and recurrent parts? That is where a trick such as proposed in Bengio et al 2016 might be useful, so that the same set of weights could be used for both.”
>
> We felt that the most natural thing to do in this work was not to tie the parameters of the feedforward and inference network. This is more in line with the amortized variational inference view - where the variational encoder network typically does not share parameters with the generative model.  However, we agree that in terms of biological plausibility, it makes more sense that the feedforward network would share parameters with the equilibrating, as in [Bengio et al 2016].  This, however, was not the aspect of biologically plausible deep learning that we aim to attack in this paper.  The big problem, as we see it, is that the brain does not use backpropagation, yets seems to train fast inference networks.  Once that is solved, we can attack the other aspects of biological plausibility.
>
> We have also added some emphasis in the Discussion section on the other motivation for this work - the design of future hardware for deep learning.  This work could be a useful starting point for the design of an efficient analog circuit for training feedforward networks.
>
> ----
> - “It might be good to mention Bengio et al 2016 in the introduction since it is the closest paper (trying to solve the same problem of using a feedforward net to approximate the true recurrent computation), rather than pushing that to the end.“
>
> We agree the this paper is very relevant, but couldn’t find it a way to work it into the introduction without having to explain the idea prematurely.  We agree that it should be more emphasized here, so we’ve moved our mention of it to the beginning of the related work section.

---

### Official Review · AnonReviewer4 · 2018-11-08
**review initialized equilibrium propagation**

**Rating:** 5
**Confidence:** 4

**Review:**

Summary:
This paper aims at improving the speed of the iterative inference procedure (during training and deployment) in energy-based models trained with Equilibrium Propagation (EP), with the requirement of avoiding backpropagation. To achieve this, the authors propose to train a feedforward network to predict a fixed point of the "equilibrating network". Gradients are approximated by local gradients only. The method is compared to standard EP on MNIST.

The overall idea of the paper to speed up the slow iterative inference (during training and deployment) seems very reasonable. However, the paper seems to be still work in progress and could be improved on the theoretical side, the presentation, and especially the experimental evaluation.
The paper is rather weak on the theoretical side. The main theoretical result is perhaps the analysis of the gradient alignment. However, I cannot follow their analysis and suspect that it is false. More detailed comments follow. Regarding the presentation, I found many typos which I don't consider in my evaluation. However, there are both minor and major issues with several equations. Details follow below. Another major concern is the lack of experimental evaluation. There is only a single plot that shows the learning curves of EP and the proposed Initialized EP with 2 different numbers of negative-phase steps and for 2 different architectures. The authors should put a lot more effort into the evaluation. For example, evaluate the influence of the hyperparameter in Eq. (10) (Is lambda > 0 detrimental to the capacity of the equilibrating network?), etc.

Lastly, as of my current understanding, the whole motivation for the EP framework is biological plausibility. In my opinion, this paper lacks a discussion of that motivation with respect to the proposed approach.

To summarize, there are too many major problems that cannot be addressed only in the rebuttal phase.


Details:
- Sec. 1.1. Equilibrium Propagation --> Sec. 2 (It is not part of the introduction)
- In 1.1., "Equilibrium Propagation is a method for training a Continuous Hopfield Network for classification". EP is a method for training various energy-based models, not just hopfield networks.
- Eq. (1): I find the notation very confusing. Specifically, I can't make sense of:
    a) "$\alpha = \{\alpha_j: j \in  S\}$ denotes the network architecture". What does it mean for alpha to denote an architecture? Please be more specific.
    b) In the definition of $\alpha_j$, you are constructing a set of neurons $i \in S  \cup I$, but then you are re-defining i in the same set, using the forall operator.
    c) Even if the two above is corrected, I can't follow. Please simplify the notation (the energy function is not that complicated).
- Eq. (1): Why is it $i \in S$ everywhere, rather than all neurons, including input neurons (as in [Scellier and Bengio 2017])?
- The text between Eq. (2) and Eq. (3) introduces the classification targets by adding the gradients of another energy function $C(s_O, y)$ to the previously described energy function from Eq. (1). First $C(s_O, y)$ is nowhere defined. Second, The energy is a scalar, while the gradient is a vector, so there must be a mistake. I suppose it should be just $C(s_O, y)$ rather than its gradients?
- Eq. (6): $f_{\phi_{j}}$ is defined as a function of multiple $f_{\phi_{i}}$ ?
- Eq. (9): Again the index i is used twice.
- Sec. 2.1: Can you elaborate on why the equilibrating network can create targets that are not achievable by the feedforward network? Is it a problem of your particular choice of model architecture? Isn't the "regularization" then detrimental to the (capacity of the) equilibrating network?
- In Sec. 2.2 on page 5, you claim that given random parameter intitialization, the gradients should almost always be aligned. For random weight matrices, where the weights are drawn with zero mean, I cannot see how this is true. To compute gradients of layer $l$, backpropagation (in an MLP) computes the matrix-vector multiplication between transposed weight matrix and the gradients of layer l+1 (I am ignoring the activation function here). The resulting gradient should have zero mean.
- Eq. (11): Is it the L1 Norm or L2?
- Eq. (12): In the preceding text, you made claims about the gradient alignment for random parameter initialization. In Eq. (12) you analyze the gradients close to the optimum?
- Eq. (12): What is f, it has never been defined. I suppose it should be the h from above?
- Eq. (12): I don't understand how you arrived at these gradient equations, even the first one. Shouldn't it be the standard backpropagation in an MLP or am I missing something? Using the chain rule $\frac{\partial L_1}{\partial w_1} = \frac{\partial L_1}{\partial s_1} \frac{\partial s_1}{\partial w_1}$, I arrive at a different result. How can there be the derivative of f (or h) twice.
- Sec. 3: Is beta really sampled from a zero-centred uniform distribution? On page 2, beta is introduced as a small positive number. Would a negative beta not cause the model to settle to a fixed point where maximally wrong targets are predicted?


[Scellier and Bengio 2017] Equilibrium Propagation: Bridging the Gap Between Energy-Based Models and Backpropagation

---

> ### Author Response · Authors · 2018-11-26
> **Response**
>
> Thank you for doing such a detailed review of our work.  We have addressed all of your points in our paper.  We have used your feedback to simplify notation, add two experiments, and better support the theoretical analysis in the text.  We hope that our changes allay your concerns, and we appreciate time and effort you put into reviewing and improving our paper.
>
> ----
>
> - “Eq. (1): Why is it $i \in S$ everywhere, rather than all neurons, including input neurons (as in [Scellier and Bengio 2017])? “
>
> The phrasing eq. 1 of in  [Scellier and Bengio 2017] is slightly imprecise, because in reality no nonlinearity is applied to the input.  We instead separate out the input terms from the other terms in the network.  This also allows us to just use “s” to describe the state of the network, and not need the additional vector “u” to describe “the state of the network AND the inputs:”.
>
> ----
>
> - “Sec. 2.1: Can you elaborate on why the equilibrating network can create targets that are not achievable by the feedforward network? Is it a problem of your particular choice of model architecture? Isn't the "regularization" then detrimental to the (capacity of the) equilibrating network? “
> - “ For example, evaluate the influence of the hyperparameter in Eq. (10) (Is lambda > 0 detrimental to the capacity of the equilibrating network?)”
>
> I rewrote this section (now numbered 2.3) to clarify why the function of the equilibrating network is more “flexible” than that of the feedforward network.  You are correct that the “regularization” is then detrimental to the capacity of the equilibrating network, but that is not important if our objective is just to use the equilibrating network to train the feedforward network.
>
> We added a new experiment in Appendix C where we sweep lambda to demonstrate its effect.
>
> ---
>
> “In Sec. 2.2 on page 5, you claim that given random parameter intitialization, the gradients should almost always be aligned. For random weight matrices, where the weights are drawn with zero mean, I cannot see how this is true. To compute gradients of layer $l$, backpropagation (in an MLP) computes the matrix-vector multiplication between transposed weight matrix and the gradients of layer l+1 (I am ignoring the activation function here). The resulting gradient should have zero mean.”
>
> The alignment effect is real, just somewhat intuitive.  It may help to think of it this way:  For a random network with parameters phi, with targets assigned by another random network with parameters phi*, the gradients d_L1/d_phi1 and d_L2/d_phi1 are indeed zero-mean random variables.  But they are not independent.  Both of these gradients depend on phi1.  Our analysis shows that when we are close to the ideal parameters (ie when phi is close to phi*), this dependency tends to produce to a positive correlation between the two gradients.
>
> To increase the reader’s confidence in this point, we have added a full derivation in Appendix B, and Figure 1 and 3, which demonstrate this alignment effect empirically.
>
> ----
>
> - “Eq. (12): I don't understand how you arrived at these gradient equations, even the first one. <….>  How can there be the derivative of f (or h) twice.”
>
> We added a full derivation in Appendix B.  The squared-derivative arises from us us doing a first-order Taylor expansion to calculate the derivative in the limit of small Delta_w.  We did, discover that we’d had an accidental a minus-sign in our equation, which we have corrected.  We have also tested them numerically (see anonymous test script at https://pastebin.com/RRgcCnrb )
>
> -----
>
> - “Sec. 3: Is beta really sampled from a zero-centred uniform distribution? On page 2, beta is introduced as a small positive number. Would a negative beta not cause the model to settle to a fixed point where maximally wrong targets are predicted?”
>
> Yes, but the learning rate is also multiplied by beta, so when it is negative it tries to raise the energy of the “maximally wrong” targets.  I have added a footnote explaining why this is done (it is a trick inherited from [Scellier & Bengio, 2017]).
>
> ----
>
> - “Lastly, as of my current understanding, the whole motivation for the EP framework is biological plausibility. In my opinion, this paper lacks a discussion of that motivation with respect to the proposed approach.”
>
> We are still many steps away from full biological plausibility.  But one of the main gaps in between Deep-Learning models and our understanding of the brain is that the brain clearly does not use backprop.  We think that addressing this issue alone (how to train a feedforward network without backprop) is sufficient scope for this paper.
>
> Moreover, there is a second motivation: Equilibrium Propagation could lead to efficient neural networks implementations in analog hardware.  However, it still requires a settling process, which is not ideal for fast inference.  We address exactly that concern.  We have added a paragraph to the Discussion making that point.

---

> > ### Comment · AnonReviewer4 · 2018-11-26
> > **greatly improved**
> >
> > Dear Authors,
> > the new version was improved greatly. Many mistakes have been corrected and most of the raised issues have been addressed.
> > The derivation in App. B is very helpful, it wasn't clear before that you were doing a Taylor approximation and hence get twice the derivative. The analysis of the regularization hyper-parameter is very useful as well.
> > I will adjust my rating.
> >
> > I do still have few issues:
> > In Eq. (3), shouldn't the first term be $E^{\beta}(s+, x, y)$ incl the loss for the target, rather than $E(s+, x)$? Then it would also be clear why beta can be negative.
> > You still use f instead of roh in a few places, e.g. after Eq. (12) and in App. B.
> > In App. B you also use both L and l.
> > In App. B, better write $g(\delta w)$ rather than g(0) in the line where you have the $lim_{\delta w \rightarrow 0}$.

---

> > > ### Author Response · Authors · 2018-11-27
> > > **Reply**
> > >
> > > Thank you for the reply and for having a sharp eye for notational errors.
> > >
> > > About Eq. (3) ... both work, and behave very similarly.  They both compute something proportional to the correct gradient in the limit of small beta.  The difference is that in the $E^{\beta}(s+, x, y)$ version parameters of the final layer directly optimise C on top the contrastive term.  The version in our paper (containing $E(s+, x)$) is consistent with the Hebbian update rule in Equation 4, so for consistency we'd like to leave it as is (also, that is the version implemented in our code).
> > >
> > > About $g(\delta w)$...We're writing out the 1st order expansion about 0 in the form "f(x) \approx f(0) + x f'(0)", and noting that in the limit of small x, this approximation becomes exact, which we believe should be correct.  We have changed our derivative notation to make clear that we are taking the "derivative of g, evaluated at 0" and not "the derivative of g(0)".
> > >
> > > We applied your other corrections in our new draft.

---

### Author Response · Authors · 2018-11-26
**Response to Reviews**

Dear Reviewers,

Thank you for providing such in depth reviews.  All of your suggestions have been used to improve our paper.  We have made the following changes in response to your comments:
- Added a derivation of the alignment effect result (code verifying the result is available here: https://pastebin.com/RRgcCnrb  )
- Changed figure 1 and added Figure 3.  These figures demonstrate on a toy problem and MNIST, respectively, that that local and distant gradients tend to align during training.
- Expanded the related work section, drawing links to variational inference and other alternative methods for neural network training.
- Simplified some notation, and clarified numerous points suggested by reviewers.

Please also see our separate responses to the comments and questions in each separate review.  We hope our improvements address your concerns.

---

### Meta-Review · Area_Chair1 · 2018-12-12
**worthwhile improvement of an existing method**

**Confidence:** 4
**Recommendation:** Accept (Poster)

**Metareview:**

The paper investigates a novel initialisation method to improve Equilibrium Propagation.  In particular, the results are convincing, but the reviewers remain with small issues here and there.

An issue with the paper is the biological plausibility of the approach.  Nonetheless publication is recommended.